# Skin Wound Healing Process and New Emerging Technologies for Skin Wound Care and Regeneration

**DOI:** 10.3390/pharmaceutics12080735

**Published:** 2020-08-05

**Authors:** Erika Maria Tottoli, Rossella Dorati, Ida Genta, Enrica Chiesa, Silvia Pisani, Bice Conti

**Affiliations:** 1Department of Drug Sciences, University of Pavia, V.le Taramelli 12, 27100 Pavia, Italy; erikamaria.tottoli01@universitadipavia.it (E.M.T.); ida.genta@unipv.it (I.G.); enrica.chiesa@unipv.it (E.C.); bice.conti@unipv.it (B.C.); 2Immunology and Transplantation Laboratory, Pediatric Hematology Oncology Unit, Department of Maternal and Children’s Health, Fondazione IRCCS Policlinico S. Matteo, 27100 Pavia, Italy; silvia.pisani01@universitadipavia.it

**Keywords:** wound healing, wound, chronic wounds, drug delivery, 3D bioprinting, electrospinning

## Abstract

Skin wound healing shows an extraordinary cellular function mechanism, unique in nature and involving the interaction of several cells, growth factors and cytokines. Physiological wound healing restores tissue integrity, but in many cases the process is limited to wound repair. Ongoing studies aim to obtain more effective wound therapies with the intention of reducing inpatient costs, providing long-term relief and effective scar healing. The main goal of this comprehensive review is to focus on the progress in wound medication and how it has evolved over the years. The main complications related to the healing process and the clinical management of chronic wounds are described in the review. Moreover, advanced treatment strategies for skin regeneration and experimental techniques for cellular engineering and skin tissue engineering are addressed. Emerging skin regeneration techniques involving scaffolds activated with growth factors, bioactive molecules and genetically modified cells are exploited to overcome wound healing technology limitations and to implement personalized therapy design.

## 1. Introduction

The skin is the largest organ in human body and plays a crucial role in different processes such as hydration, protection from chemicals and pathogens, vitamin D synthesis initialization, excretion and thermal regulation. Severe skin damage can therefore be life-threatening. Healing a skin wound shows an extraordinary mechanism of cellular function that is distinctive in nature. The repair process includes the interaction of cells, growth factors and cytokines involved in closing the lesion. The inconveniences caused by injuries, particularly for chronic wounds, are mainly related to treatment and management procedures limiting the wound repair, rather than tissue integrity restoration (so-called “restitutio ad integrum”) [1]. For this reason, several studies are oriented towards achieving more effective wound therapies, in order to reduce health costs and provide long-term relief and, ultimately, effective scar healing. Skin wound therapies are classified as “Conventional” or “Regenerative”. Conventional therapy leads to the formation of scars irrespective of aesthetic and possible functional alterations [2]. Regenerative wound therapy is a new and rapidly developing area in biomedical research; it aims to restore skin to its pristine function, reestablishing damaged cells and skin tissue without scarring [3]. In any case, regeneration strategies should be considered complementary to essential conventional treatments, such as debridement. This review focuses on the wound healing process, providing a deep description and analysis of wound healing mechanisms.

Wound medications evolving from simple “coating” devices to advanced medications, as well as their advantages and limitations in the skin regeneration process, will be investigated. In particular, this review is focused on the progress in wound medication and its development in recent years. The main complications related to the healing process and the clinical management of chronic wounds are described in the review. Moreover, advanced treatment strategies for skin regeneration and experimental techniques for cellular engineering and skin tissue engineering are addressed. Emerging skin regeneration techniques involving scaffolds activated with growth factors, bioactive molecules and genetically modified cells are exploited to overcome wound healing technology limitations and to implement personalized therapy design.

## 2. Phases and Complications of the Wound Healing Process

### 2.1. An Overview of Wounds and Their Consequences

Wounds have a variety of causes, namely surgery, injuries, extrinsic factors (e.g., pressure, burns and cuts), or pathologic conditions such as diabetes or vascular diseases. These types of damage are classified into acute or chronic wounds depending on their underlying causes and consequences [4]. Acute wounds usually proceed through an organized and appropriate repair process, resulting in the sustained restoration of anatomical and functional integrity. On the contrary, chronic wounds are not able to achieve optimal anatomical and functional integrity. Healing is related to and determined by both pathological processes nature, degree and status of host and environment. Systemic factors such as patient age, the presence of vascular, metabolic and autoimmune diseases, as well as ongoing drug therapy, may affect the wound healing process [5]. An ideally healed wound is an area returned to normal anatomical structure, function and appearance after an injury; a minimally healed wound is characterized by the restoration of anatomical continuity, but without sustained functional results; therefore, the wound can recur. Between these two conditions, an acceptably healed wound is characterized by the restoration of sustained functional and anatomical continuity.

Wound extent evaluation and classification can be performed by non-invasive and invasive technologies. Non-invasive evaluation includes the determination of wound perimeter, maximum length and width dimensions, surface area, volume, amount of weakening, and tissue viability. Invasive methods quantify the wound extent in terms of tissue levels from its surface to its depth [5]. A wound can be further described by various attributes, including blood flow, oxygen, infection, edema, inflammation, repetitive trauma and/or insult, innervation, wound metabolism, nutrition, previous injury handling, and systemic factors. All these attributes can provide evidence of the origin, pathophysiology and condition of a wound [6].

Ultimately, wounds should be assessed by considering their effect on the host, as patient status is essential in understanding the impact of systemic factors on the wound. The evaluation of the healing process is quite challenging since it is a dynamic process and it requires constant, systematic and consistent evaluation, involving a continuous reassessment of wound extent, type and severity.

Quality of life is impaired by wound persistence and the care cost manifests from both a psychological point of view and in the prolonged hospitalization time, as well as morbidity and even mortality. For these reasons, wounds have been called a “Silent Epidemic” [7]. Most of the financial costs relate to health care personnel employment, the time and cost of hospitalization and the choice of materials and treatments. For all these reasons, the development of new technologies, intended to improve the healing process, is challenging [8].

### 2.2. Phases of Wound Healing

Skin epithelial cells are labile elements that are continuously eliminated in the stratum corneum through the keratinocyte desquamation process and are replaced, in the basal layer, by differentiated elements derived from stem cell proliferation and differentiation. Cell renewal varies according to different factors, such as trauma, hormonal influences, skin conditions and individual wellbeing. However, the cutaneous regenerative process, in reference to a wound lesion, is inversely proportional to the evolution of the considered species [9]. It consists of numerous phases activated by intra and intercellular biochemical pathways and coordinated in a harmonious way to restore tissue integrity and homeostasis. Cellular elements such as the coagulation cascade and inflammatory pathways are also involved. Several cells are involved such as fibroblasts, keratinocytes and endothelial cells, and neutrophils, monocytes, macrophages, lymphocytes and dendritic cells as immune components [10]. In Figure 1, a schematic representation of the wound healing process is presented, with the cells involved in each phase.

The regeneration process involves sequential phases regulated by gene expression, via autocrine or paracrine mechanisms. The ending of active processes is achieved by gene silencing during the progression of the regeneration process [11]. Wound healing is one of most complex process in the human body, since it involves the spatial and temporal synchronization of the inflammatory phase with tissue regeneration and remodeling. The inflammatory phase follows the injurious event and it includes the coagulation cascade, inflammatory pathway and immune system involvement [12]. All these events take place to prevent an excessive loss of blood, fluids and the development of infections, and to facilitate the removal of dead or devitalized tissue. Hemostasis is achieved by platelet clot generation, followed by fibrin matrix formation, which acts as a scaffold for cell infiltration. As a result of platelet degranulation, the release of chemotactic signals by necrotic tissues, and of bacterial degradation products, the complementary system is activated and neutrophils arrive at the lesion [13]. Finally, macrophages coordinate all events evolved in response to damage. These cells are responsible for fibrin phagocytosis activity and cellular debris, and they secrete macrophage-derived growth factor (MDGF) for fibroblasts and endothelial cells [14]. New tissue formation begins within two to ten days after the lesion and consists of cell proliferation and the migration of different cytotypes. When the lesion involves the dermis, a poorly differentiated and highly vascularized connective tissue called granulation tissue is formed, which consists of cellular and fibrillar components integrated in an apparently amorphous matrix. The cells of granulation tissue are (i) fibroblasts, responsible for the synthesis of the fibrillar component; (ii) myofibroblasts, involved in the wound contraction mechanism and (iii) endothelial cells, responsible for the neo-angiogenesis process [15].

The re-epithelization process, characterized by the proliferation and migration of keratinocytes towards the core part of the lesion, originates in this phase as the area between the bottom and the edges of the wound is filled with granulation tissue. This represents the matrix in which keratinocytes, residing on lesion edges, migrate and proliferate [14]. Skin re-epithelization structural organization can be explained by two models: sliding and rolling models. According to the sliding model, keratinocytes of the basal layer suffer a modification of their anchoring joints (desmosomes and hemidesmosomes), allowing their detachment and lateral migration into the core part of the lesion. According to the rolling model, keratinocytes go through a morphological and functional modification, together with desmosomes, resulting in them rolling towards basal keratinocytes, which instead remain anchored to the basal membrane [16]. Basal layer regeneration leads keratinocytes to proliferate and differentiate vertically, restoring the physiological features of the multilayered epithelial tissue.

The remodeling phase starts about three weeks after an injurious advent and lasts for over a year. During this phase, all processes activated in previous phases are silenced and macrophages, isolated endothelial cells and myofibroblasts run into apoptosis or they are relocated from the wound, leaving a region rich in collagen and other extracellular matrix deposition (ECM) proteins. Interactions between the epidermis and dermis, together with additional feedback, allows the continuous regulation of skin integrity and homeostasis. Type III collagen, located in ECM, is gradually replaced in 6–12 months [17].

### 2.3. Acute and Chronic Wound Healing

Acute wounds (such as traumatic and surgical wounds) pass through the normal wound healing stages, resulting in an expectable and organized tissue repair arrangement [18]. On the contrary, chronic wounds involve a disordered repair process and they can be mainly classified into vascular ulcers (such as venous and arterial ulcers), diabetic ulcers and pressure ulcers [19]. Chronic wounds exhibit a persistent inflammation phase, resulting in microorganism recruitment, biofilm development [20], and the release of platelet-derived factors, such as transforming growth factor-beta (TGF-β), or ECM fragment molecules. The pro-inflammatory cytokine cascade, including interleukin-1β (IL-1β) and tumor necrosis factor α (TNFα), continues for a prolonged period, leading to important levels of protease in the wound bed. In chronic wounds, protease levels go above those of inhibitors, triggering ECM destruction, and boosting proliferative and inflammatory phases [21]. Inflammatory cells collected in the chronic wound bed rise to levels of reactive oxygen species (ROS), resulting in ECM protein injuries and premature cellular senescence [22]. Chronic injuries are also characterized by phenotypic defects in the cells and dermis, such as reduced growth factor receptor density and mitogen potential, inhibiting resident cells from responding adequately to wound healing signals [23,24,25] On the contrary, proteases are tightly regulated by their inhibitors in acute injuries, avoiding ECM destruction and supporting the proliferation phase.

### 2.4. Inside a Chronic Wound: Physical Elements of the Chronic Wound Healing Process

The main physical manifestations of chronic wounds are represented by exudation, persistent infection and necrosis, which are responsible for wound management and care complexity [26].

#### 2.4.1. Exudate

The wound exudate is a reflection of wound bed physiology. Similarly, the skin wound creates an exudate that represents the micro-environment of the insulted tissue. The exudate is a marker of the chronic state of an injury or a sign of wound treatment effectiveness. There is increasing evidence that destructive effects observed in chronic injuries can be aggravated by exudate components that, being corrosive in nature, result in continuous ECM degradation. The isolation of these components has identified metalloproteinases (MMPs), in particular MMP-9s, as dominant components in the destructive process; moreover, a relationship between elevated bacterial and MMP-9 levels in chronic wounds has been established [27]. In addition, exudation may be the first indicator of possible systemic complications [28]; the signaling of mediators and protein content can provide information about the type of tissue involved in the damage and facilitate the selection of the most appropriate treatment approach.

#### 2.4.2. Infection

After an injury, the skin activates an inflammatory mechanism that not only produces exudate, but also leads to the formation of antimicrobial peptides (AMPs) in response to infection. AMPs are amphipathic peptides and they are constitutively expressed or induced after cellular activation in response to inflammatory or homeostatic stimulation. The most carefully studied AMP families in the human skin are defensins and cathelicidins, which are produced by a variety of skin cells such as keratinocytes, fibroblasts, dendritic cells, monocytes, macrophages and sweat and sebaceous glands. The access of bacteria in a skin region that has suffered an insult is an inevitable phenomenon and sometimes the immune action turns out to be ineffective, leading to complications and even deaths in subjects with important chronic skin lesions [29]. Healthy skin is richly populated by bacteria that play an important role in skin ecosystem. In the case of the interruption of skin continuity, the bacteria migrate from the skin surface to regions in which they are not normally hosted, causing an imbalance that leads to infection in the skin wound. Bacteria may originate from the external environment, such as *Staphylococcus aureus*, or from bacteria residing in hollow organs migrating through the blood pathway. An additional bacterial risk is represented by biofilm formation, a micro-environment layer rich in glycoprotein that adheres to the wound bed, protecting bacteria and enhancing their proliferation. The biofilm matrix makes bacteria tolerant to challenging conditions and resistant to antibacterial treatments. In addition, biofilms are responsible for causing a wide range of chronic diseases and, due to the emergent antibiotic resistance in bacteria, it has become very difficult to effectively treat them [30,31].

#### 2.4.3. Necrosis

Devitalization or/and necrosis arise when an infection is unresolved, or the tissue has irreparable damage. Frequently, superficial infections progress towards deep tissue layers, involving bone tissue and sometimes affecting systemic pathways, leading to generalized sepsis and bacteremia. Skin necrosis is characterized by a wide range of etiologies including external factors or, more frequently, vascular occlusion. Necrosis is a serious disease defined as the death of cells or tissue for pathological reasons; usually, it shows up as a purplish, bluish or black skin coloration, and it is irreversible. When it is accompanied by bacterial infection and decomposition, gangrene is mentioned. The main necrotizing infections are as follows: ecthyma, a bacterial infection that causes ulcerations and scabs; necrotizing fasciitis, an infection that causes rapid necrosis of subcutaneous fat with production of malodorous serum; and acute meningococcemia, which causes an acute petechial eruption that can be followed by ecchymosis and ischemic necrosis [32].

### 2.5. Difference between Healed and Physiological Tissue: Scarring

The human body always reacts to an injury, activating the wound healing process and scar formation. Scars are efficient neo-formation tissues, however they do not reproduce characteristics and functions of physiological tissue that they replace [33]. Any damage in humans is repaired by a neo-formation that replaces the missing tissue with an extracellular matrix, consisting mainly of fibronectin and collagen types I and III, and there are some skin components that will not recover after a serious injury, such as subepidermal appendages, hair follicles or glands [34]. The scar tissue matrix, represented by granulation tissue, is the final product and it is characterized by a high density of fibroblasts, granulocytes, macrophages, capillaries and collagen fibers [35]. In the primordial scar tissue phase, angiogenesis is not yet complete, although it is abundantly present and it appears reddened. The dominant cells at this stage are fibroblasts, which have different functions such as collagen production and ECM components (e.g., fibronectin, glycosaminoglycans, proteoglycans and hyaluronic acid (HA)). At the end of this phase, the amount of fibroblasts in the maturation phase is reduced by their differentiation in myofibroblasts [36]. Scar formation ends in the remodeling phase of the wound healing process (Figure 2), it starts at day 21 and goes on for 1 year after injury. During wound maturation, ECM components undergo constant changes. Collagen III, which is produced in the proliferative phase, is now replaced by the strongest type I form of collagen, which is oriented in small parallel bundles, differing from the healthy dermis texture [34]. Then, myofibroblasts cause the contraction of the wound due to their strong adhesion to collagen, helping with wound healing. In addition, angiogenic processes and blood flow in the wound bed decrease, acute wound metabolic activity slows down and eventually it stops, leading to mature scar formation. Scar formation is the physiological endpoint of wound repair in mammals. When excessive scarring occurs, there is an imbalance between biosynthesis and degradation, mediated by apoptosis and ECM degradation, and this dysfunction leads to a persistent inflammatory phase, a prolonged proliferation phase and reduced remodeling [32]. Hypertrophic scars contain excessive microvessels, which are mostly occluded due to the over-proliferation and functional regression of endothelial cells, induced by myo-fibroblastic hyperactivity and excessive collagen production. Changes in ECM and the epithelium also appear to be involved in abnormal scarring; mechanical stress stimulates skin mechanical–sensory nociceptors, which release neuropeptides involved in vessel modification and fibroblast activation [5].

## 3. Chronic Wound Care Treatment Approaches

### 3.1. Wound Care through TIME Principle

In the practice of wound care, the acronym Tissue (T), Infection (I), Moisture (M) and Epithelial (E) (“TIME”) synthesizes all main factors that interfere negatively in the healing process. There is no systematic way to assess acute injuries; therefore, TIME can be used as a practical guide for chronic wound management [37]. In the acronym TIME are described some fundamental concepts that can be grouped into four areas [38]: Tissue (T): evaluation and debridement of devitalized or non self-material in the wound bed (including necrotic tissue, adhering dressing material, biofilm or debris related to multiple organisms). Infection (I): assessment of etiology and treatments for infection management using systemic or topical antibiotics. Moisture balance (M): evaluation of etiology and management of wound exudate. Epithelial edge advancement: evaluation of the progress of edges and surrounding skin status [23]. Knowledge of the TIME concept and the molecular biology of injuries has led to new developments in chronic wound management treatments and technologies as detailed below (Figure 3).

#### 3.1.1. Tissue (T): Debridement Procedure

Over the past decade, new therapies for wound debridement have been developed, such as low-frequency ultrasounds, hydro-surgery devices, larvae and enzyme agents. It is widely recognized that devitalized tissue removal is a fundamental process for tissue repair [39]. Debridement is optimal for preparing the wound bed, but in some cases is not recommended (i.e., in immunosuppressed patients). Once all injury assessment factors have been revised, and wound debridement has been decided as an appropriate option, the most appropriate debridement method should be selected [40], taking in consideration the amount of exudate produced by the wound. The most common debridement types are as follows. Autolytic: the body uses endogenous enzymes and moisture gradients to remove devitalized and necrotic tissue. This is a long process, more suitable for minor injuries. Wound care products such as hydrogels, films, honey and hydrocolloids can be used to support this natural process and allow wound healing in a humid environment [39]. Surgical: surgical removal allows us to identify the entire condition of the wound. Sterile instruments are used to remove devitalized or necrotic tissues; often, this procedure also removes some vital tissue. Surgical debridement is fast, safe, minimizes the risk of infection and chronic wound complications [41]. Mechanical: this is an effective and economical method of debridement. The disadvantages of this method are the lack of selectivity and pain. It can be applied through hydrotherapy, using water jets to wash residues from the wound surface. Another mechanical debridement method is wet–dry therapy, where a wet gauze is applied to a wound and then left to dry; once dried, the gauze that bound the necrotic or devitalized tissue is removed from the wound bed [42]. Biological: the application of sterile larvae-secreting enzymes, able to liquefy dead tissue; the liquid-containing bacteria is then ingested and neutralized by larval bowels. In addition, there is an increase in wound bed growth rates and advantageous changes to skin pH values. This procedure provides a safe and selective debridement method; however, it is poorly accepted by patients and doctors [43]. Enzymatic: this procedure involves the application of enzymes into the wound bed, with a proteolytic action on the necrotic tissue. It is a pH-dependent process and proteolytic enzymes can be deactivated by specific agents. Some examples of proteolytic enzymes are papain, collagenase and fibrinolysis [44].

#### 3.1.2. Infection (I): Prevention Strategies

After the proper cleaning of the wound bed and a debridement procedure, an antiseptic product is needed to prevent infection and biofilm formation. Many antiseptic products are cited in the literature and their selection should consider toxicity towards granulation tissue. Antiseptics are chemical products, capable of preventing or stopping the action of microorganisms, either by inhibiting their functions, or by destroying them [42]. They are recommended only when unavoidable. The biotechnology industry has developed substances with local and selective antiseptic action and limited side effects in tissues, such as local slow-release antiseptics, i.e., substances exploiting their action only when released at the site of injury, with doses selectively more effective on bacteria and less effective on tissues. Common antiseptics include iodine, silver (including silver sulfadiazine, which has a potent bacteriostatic action), polyhexanide and betaine (PHMB), sodium hypochlorite, chlorhexidine and acetic acid. Iodine powder can be applied to the wound site; the powder allows the absorption of exudate and creates an environment less favorable for bacterial proliferation. Silver, which is one of the historical antibacterial agents, unlike antibiotic molecules, does not induce resistance in bacteria and, for this reason, it is a useful treatment [45]. Silver sulfadiazine 1% causes an ultrastructural change in the bacterial membrane and it destroys bacterial DNA through irreversible bonds inhibiting bacterial cell respiration and altering electrolyte transport and folate production. Silver sulfadiazine 1% is an efficient treatment in wound management and is on the market as a polyurethane foam that acts by occluding the wound bed and guaranteeing its transpiration. This device allows the absorption of exceeding exudate and create a microenvironment unfavorable for bacterial proliferation, delivering silver. There are also absorbent gauzes with activated carbon (AC) and silver cores (SCs), combining AC absorption properties with the bacteriostatic action of silver. New products proposed by pharmaceutical companies are breathable and non-adherent gauze, soaked in a bacteriostatic substance such as honey, which has therapeutic properties. Ultrasound is another method exploited for fighting infections, it separates devitalized tissues from healthy ones through the mechanism of cavitation [46]. Furthermore, this treatment alters the bacterial membrane, increasing its permeability to antibacterial substances. All the main guidelines agree in recommending the non-use of local antibiotics, because more disadvantages than advantages have been highlighted. Local hypersensitivity reactions or dermatitis are examples of adverse effects due to topical use of antibiotics [47].

#### 3.1.3. Moisture (M) Balance and Exudate Management

There are differences between the exudate composition of acute and chronic wounds. The exudate of acute wounds is rich in leukocytes and nutrients, while that of chronic wounds has high levels of proteases, pro-inflammatory cytokines and MMPs. The increased proteolytic activity of chronic exudate inhibits healing, damages the wound bed, degrades the extracellular matrix and destroys skin integrity. Additionally, high levels of cytokines promote and prolong chronic inflammatory responses. For this reason, an adequate equilibrium of wound moisture is required. Too much exudate causes an injury to the surrounding skin; low amounts of fluid inhibit cellular activities and lead to formation of scar tissue. Bad exudate management could lead to biofilm formation. Therefore, exudate volume and viscosity should be considered during dressing screening. Most common treatments for exudate management are absorbent medications and negative pressure wound therapy (NPWT). Some types of absorbent dressings include films, hydrogels, acrylics, hydrocolloids, calcium alginates, hydrofibers and foams [48]. The composition of hydrogel formulations includes insoluble copolymers capable of binding water molecules. Water present in the matrix can transfer to the wound, whereas the matrix itself is able to absorb the wound exudates, maintaining an optimal level of moisture [46]. Alginate-based products (calcium alginate, sodium alginate or alginic acid) are hydrogels able to absorb wound exudates and maintain a moist wound environment [48]. Traditional dressings are impermeable to water vapor, difficult to remove and show a poor absorption capacity; advanced medications overcome these drawbacks. One example of such advanced medications would be hydrofibers, which are dressings composed of carboxymethylcellulose sheets with a high absorbance capacity and a simple removal procedure.

#### 3.1.4. Epithelial (E) Edge Advancement

The evaluation of wound edges may indicate whether the contraction of wounds and epithelialization is progressing, providing essential signs about treatment effectiveness or the need for reassessment. Treatments addressed to improve wound healing and wound edge advancement include electromagnetic therapy (EMT), laser therapy, systemic oxygen therapy and negative pressure wound therapy (NPWT). Pulsed EMT consists of a short-term energy emission that has the advantage of protecting tissues from damage and heat generated by continuous emissions. However, further studies are needed to explore the effects of EMT [49,50,51,52,53]. In conclusion, complete and timely wound closure is the main objective of all aspects of wound care, although this is not always achievable. Chronic wounds, in particular, are challenging to effectively treat. Most clinical guidelines are still a work in progress because of the continuous improvements in wound pathology, healing and therapeutic agents. Although the basic principles of TIME have not changed greatly since its first inception, its applications have been expanded with advances in knowledge and wound management [38].

### 3.2. Advanced Dressings: The Transition of Advanced Dressings from Passive to Active Role in Wound Healing Process

Advanced dressings (AD) are composed of different materials that can facilitate the healing process in all its phases. These new products are often able to remain active on the wound bed for several days, reducing the number of medications and procedures needed for their replacement. Advanced medications contribute significantly to the healing of complex wounds (such as chronic wounds), reducing health care costs. In general, a dressing is a device used to remedy damage caused by an injurious stimulus, providing protection to the wound from the external environment, promoting healing and reducing the risk of infection. There are different types of medication and each of these has a specific purpose. Prevention: dressings for preventing injuries that could occur in areas under pressure, for example, in people who are forced to remain motionless for a long time. Coatings: called second dressings, such as ointments, creams, etc. Protection: dressings for protecting the wound bed from external contamination or mechanical trauma, to avoid alterations in the healing process, for example, in the case of surgical wounds. Cure: aimed at promoting healing, as well as protecting against exogenous agents and trauma.

Medications performing prevention and coating actions are relatively simple and can be classified as “Traditional dressings” (TDs), whereas those addressing protection and cure are classified as “Advanced dressings” (ADs). Simple medications by TDs are performed on minor lesions, which show minimal secretions and tend to promote rapid healing; some examples are slight surgical incisions, erythema or non-severe ulcers. The dressing is applied on the skin to protect the wound, but it does not play an active role in the healing process [54]. These dressing are characterized by their low cost and ease of use. However, TDs have many disadvantages, including the promotion of ischemia/necrosis and the need for frequent substitutions. In order to overcome these disadvantages, studies have been carried out on the formulation of innovative wound care technologies; this has led to the development of ADs that play an active role in treating more serious injuries with complex healing processes [55]. These types of medications exploit the properties of biocompatible materials of natural or synthetic origin and stimulate, through interaction with tissues, a response aimed at faster healing [56].

The roles played by biomaterials used to make ADs are as follows. Active: materials that play an active role in wound treatment and healing stimulation; passive: compounds that absorb exudation and protect the wound from external agents; interactive: agents that control the lesion microenvironment, managing the healing process. Scientific progress in ADs has led to the consideration of a wound dressing as a medication that is completely involved in the healing process.

Regardless of the nature of the lesion and the method used, an ideal AD should be biocompatible and biodegradable, and should have optimal water adsorption and retention properties, low cytotoxicity, nonstick ability, and antibacterial effects. It should allow gas exchange, and trap wound exudates to maintain the hydration of the wound. ADs could act as platforms for the delivery of different active agents (such as growth factors, anti-fibrotic, anti-microbial, and anti-inflammatory agents, small-molecule drugs, nucleic acids) or cells that boost the synergy of wound healing [37].

Currently, the human recombinants platelet-derived growth factor (PDGF), fibroblast growth factor (FGF) and epidermal growth factor (EGF), are extensively investigated for wound repair application [8]. However, their low absorption capacity, short in vivo half-life and high risk of carcinogenesis restrain their use in wound repair.

Cutisorb™, Iodosorb, and Actisorb Silver 220 are few examples of antimicrobial dressings already on the market. Acticoat and Acticoat medications are a new generation of product; they exploit a new nanocrystal silver coating technology and the dressings are properly designed for preventing adhesion to the wound surface, controlling bacterial growth and promoting wound treatment. These ADs are recommended for treating partial- or full-thickness wounds (pressure, venous, diabetic and chronic wounds and leg ulcers) where infections are very common [37]. In Appendix A, we present a comparative summary of emergent skin wound care and regeneration technologies.

## 4. Skin Regeneration Process and Skin Regeneration Therapies

### 4.1. Difference between Wound Healing and Regeneration Process

The use of skin grafts is needed for replacing surface deficits that cannot be resolved by a simple approximation of wound margins. Grafts are invasive procedures that can expose the patient to serious complications; for this reason, they are carried out only when there are no alternatives such as ADs [57]. The challenge to overcome the current barriers associated with wound care requires innovative management techniques, such as regenerative medicine. Regenerative medicine is a new area of medical science, aimed at improving the regeneration process through a multidisciplinary approach focused both on problem solving through a reparative approach, and on remedying deficiencies related to the physiological process of wound healing [58]. In less phylogenetically evolved animals, regeneration is a physiological process; many larval and adult animals are able to regenerate large sections of their body plan after transection or amputation [59]; unfortunately, in humans, this occurs only during the first part of intrauterine life.

Regenerative medicine studies provide a number of opportunities to accelerate and promote wound healing. Growth factors, stem cells and biomaterials can be applied directly to induce regeneration or indirectly change the wound environment and stimulate healing. This multidisciplinary approach opens up future perspectives for tissue regeneration. Collaboration is fundamental to connect clinicians with scientific engineering skills to commercial teams and to guide new technologies towards a safe and effective implementation [60]. The criteria proposed by all different disciplines can be analyzed individually, and must then be clustered and clinically evaluated in the interest of combining patient needs with the available technologies. Safety is a clear priority in clinical practice and a best-fit risk analysis must include attention to diseases and treatments offered by regenerative medicine. The site of lesion the should be the target for aesthetic and functional considerations, considering the broad variability of the skin in different body districts. The timely clinical availability of these treatments is a key factor, especially for acute damage or injuries that endanger a patient’s life. The economic factor is also an important element; high-quality outcomes are crucial to justify the costs of new technologies. Finally, it should be considered that, now, there is no complete solution for skin regeneration, given its structural and functional complexity. However, collaboration among different disciplines provides a real opportunity to improve the clinical care of difficult wounds.

### 4.2. Therapeutic Potential of Regenerative Medicine in Wound Healing

Key components of regenerative medicine, such as growth factors, autologous cells and stem cells, gene therapy and tissue engineering, can be used to address different stages of wound healing. Several approaches, such as angiogenesis, immune modulation, cell proliferation and extracellular matrix deposition (ECM) can be exploited to induce regeneration.

#### 4.2.1. Growth Factors Involved in Stimulating Wound Healing

The tissue repair process is controlled by the interaction of growth factors with specific cell surface receptors; these interactions stimulate cell migration, trigger angiogenesis, epithelialization, and encourage matrix formation and the remodeling of the injured site [61]. Several growth factor families were investigated for wound healing, such as epidermal growth factor (EGF), fibroblast growth factor (FGF), transforming growth factor beta (TGFβ), and platelet-derived growth factor (PDGF). There is also emerging evidence of the role of stromal cell-derived factor 1 (SDF-1) in regulating epidermal cell migration and proliferation during wound repair [58].

EGF is secreted by platelets, macrophages and fibroblasts and it plays an important role in epithelialization. FGF is produced by keratinocytes, mast cells, fibroblasts, endothelial cells, smooth muscle cells and chondrocytes; it promotes granulation tissue formation, epithelialization and matrix formation [62]. Platelets, keratinocytes, macrophages, lymphocytes and fibroblasts produce TGFβ, which is crucial in inflammation, granulation tissue formation, epithelialization, and matrix formation and remodeling. PDGF is produced by platelets, keratinocytes, macrophages, endothelial cells and fibroblasts and also plays a role in each stage of wound healing [63].

Pierce et al. revealed that both PDGF and TGFβ accelerated in vivo wound repair, but through specific mechanisms. Briefly, PDGF is involved in macrophages and fibroblasts’ chemo-attraction and stimulates them to express growth factors, including TGFβ [64]. The role of each growth factor in wound repair has been proven by several researchers; in addition, some studies verified the potential of using growth factors in combination and carriers for their delivery to maximize wound healing [58]. Although the topical application of growth factors was shown to accelerate wound healing, there are barriers to their therapeutic application. These factors undergo rapid degradation from proteolytic factors; many studies aim to find the correct combination of biomaterials and growth factors in order to formulate a suitable carrier for preserving growth factor integrity and stability. As wound repair is a dynamic process, it remains to be clarified whether the provision of growth factors should be sustained or transient, and how long they are required for an extensive repairing. Furthermore, there is much interplay between cells and components of the wound healing cascade. The challenge is to combine different growth factors and relate them to a specific injury environment. A dynamic environment, such as the one occurring in the naturally driven wound healing process, could be a promising approach to ensure an effective combination treatment [61].

#### 4.2.2. Cellular Skin Substitutes

Cellular skin substitutes have shown great potential by providing all the elements needed for skin regeneration, such as cells, mediators and materials mimicking ECM [65]. Viable cells should be cultured in special conditions in order to prevent ECM damage during cell sheet substitute production, and treatment with trypsin should be avoided [66]. Human skin cells, fibroblasts and keratinocytes are primary sources of dermal substitute production. Fibroblasts are dermal cells and are responsible for the synthesis of most of ECM structural components (e.g., collagen, elastin, laminin and glycosaminoglycan). As for keratinocytes, they are cellular components of the epidermis and look different depending on the stage of their maturation process; in addition, they play a protective function, representing an effective defensive barrier against the external environment. Therefore, specific cell composition constructs have been developed according to the treatment target, such as engineered constructs based on keratinocytes, intended for epidermal regeneration or based on fibroblasts, for dermal regeneration. In addition, fibroblasts and keratinocytes communicate with each other through a paracrine crosstalk that leads to cell recruitment, which is required for complete wound closure. For this purpose, double-layer dermal cellular substitutes have been produced, containing both fibroblasts and keratinocytes, for the repair and regeneration of skin-deep wounds. When applied to the wound site, cells provide signal molecules, growth factors, and extracellular matrix proteins supporting skin tissue regeneration. Commercially available products incorporate keratinocytes (e.g., Epicel), fibroblasts (e.g., Dermagraft^®^) or both keratinocytes and fibroblasts (e.g., Apligraf^®^) [14]. The novelty of these products is their capability to promote skin regeneration as a function of their structure and composition. The most used matrix is based on collagen, which is totally biocompatible and biodegradable. In most cases, collagen comes from carefully selected cattle (to avoid bovine spongiform encephalopathy). After being derived from bovine tendons, collagen undergoes processes of purification from cells, DNA, RNA and all proteins, with the exception of glycosaminoglycan (GAG) which allows its long-term permanence in the wound bed; moreover, GAGs help to contain inflammation and to maintain a proper osmotic gradient. Subsequently, the purified collagen undergoes changes that involve immunogenic telopeptide deletion. Atelocollagen is important for many pro-regenerative actions; it acts as a real biological modulator of the environment when injected, favoring regenerative processes. The scaffold should be assembled according to its specific conformation and structure; in particular, pore sizes and their distribution are essential to providing a suitable matrix for effective cell migration and arrangement. These scaffolds represent a basis for revascularization, forming a proper microenvironment for cell migration and proliferation [67]. An example of a complete skin replacement is Apligraf, a bi-layered bioengineered skin substitute and the first engineered skin substitute approved by the US Food and Drug Administration (FDA) to promote the healing of ulcers that have failed standard wound care. It is obtained by stratifying human foreskin-derived neonatal fibroblasts in a bovine type I collagen matrix and human foreskin-derived neonatal epidermal keratinocytes. Apligraf provides both cells and a matrix for non-healing wounds, producing cytokines and growth factors similarly to healthy human skin; however, its exact mechanism of action is still unknown [68]. In addition, these devices can be equipped with covering components consisting of breathable material to protect the skin during the regeneration process; it should be borne in mind that cellular skin substitutes cannot be applied to all chronic injuries as special wound bed conditions are needed to achieve optimal effectiveness.

The high risk of morbidity along with limits in donor sites restrict the use of autologous skin cells for extensive wound healing. In this context, stem cells (SC) are a promising alternative since they have a self-renewal capacity, multi-lineage differentiation potential and they can be retrieved from several tissues, such as embryonic, fetal and adult tissues. Epidermal stem cells (EpSC) and their progenitors could be an attractive source in wound therapies since they are included in the epidermal basal layer and terminal hair follicles. EpSCs and related progenitors could be considered sources of autologous cells for chronic wounds [69]. Adipose-derived stem cells (ADSCs) and adipocytes were widely investigated in wound healing; Kim and colleagues investigated the wound healing effect of ADSCs, both in vitro and in vivo, on acute wounds [70]. In vitro testing proved ADSCs with the ability to promote the proliferation and migration of human dermal fibroblasts (HDFs) through cell-to-cell interactions and paracrine activation mediated by secretory factors Moreover, an in vivo experiment on nude mice presented a significant reduction in wound size and rapid epithelialization from the wound edges after 7 days after ADSC treatment.

Their safety and easy isolation procedures make MSCs a great alternative for wound regeneration; MSCs retrieved from skin, fat and bone marrow have shown promising evidence of accelerating the healing process in both acute and chronic wounds [58]. Moreover, MSCs allow the regeneration of skin appendages, such as hair follicles, sweat glands and microvessels. MSCs speed up wound healing processes and increase healing outcome quality through direct differentiation and cell paracrine signaling [71]. They increase angiogenesis, modulate inflammatory response and scarring [72]; wound closure is promoted by accelerating the recruitment of macrophages and endothelial cells mediated by pro-angiogenic cytokine production (vascular endothelial growth factor (VEGF), hepatocyte growth factor (HGF) and fibroblasts and fibroblast growth factor (bFGF)), as well as the migration of fibroblasts and keratinocytes into wound bed. The Rigenera protocol is an example of the potential provided by SCs, specifically MSCs and their progenitors, as this protocol is used to fragment autologous connective tissue in several species and to select specific cell populations including MSCs and their progenitors. Cells preserve their differentiation capability, and they stimulate the activity of quiescent SC niches located into and surrounding the wound bed, regenerating impaired tissue [73].

#### 4.2.3. Gene Therapy

Gene therapy uses genes for treating or preventing diseases. In the not too distant future, several diseases will be cured by inserting specific genes into patients’ cells; several approaches have already been exploited, including replacing mutated genes with healthy genes; inactivating, or knocking out, mutated genes and introducing into the body a new gene. Gene therapy is a promising treatment option for a number of diseases (including inherited disorders, some types of cancer, and certain viral infections); however, it is risky because of its poor safety and effectiveness [74].

Mavilio et al., 2006, used epithelial stem cells, known as holoclones, in a patient with junctional epidermolysis bullosa (JEB). JEB is caused by mutations in genes coding for the basal membrane component in laminin 5 (LAM5); these gene mutations are devastating and often result in fatal skin adhesion disorder. Small pieces of epidermis were isolated from patients and treated by a normal version of LAMβ3 using a retroviral vector. The vector integrated into each cell’s genome, enabling normal LAMβ3 expression. The in vitro genetically corrected cells formed a larger piece of epidermis that was transplanted onto the patient’s leg [75]. Hirsch and colleagues exploited this strategy on a seven-year-old child who had an extremely severe form of epidermolysis bullosa caused by LAMβ3 mutations. Grafts of about 0.85 m^2^ were implanted into the patient with full recovery after 21 months [76].

Nevertheless, the vector integrates into the host’s genome in random sites and this could interrupt the expression of essential genes or overexpress genes that control tumor development. To investigate this possibility, Hirsch and colleagues sequenced the patient’s DNA, revealing that most of the integrations occurred in non-coding sequences, demonstrating the safety of the treatment. Moreover, this treatment could be more effective in children, whose stem cells undergo higher renewal. Technologies such as Clustered Regularly Interspaced Short Palindromic Repeats/Cas9 (CRISPR’Cas9) are essential strategies to correct certain mutations. Indeed, stem cell and gene therapies are often considered to be the future of medicine, but further studies are needed to broaden these types of strategies in common clinical practice [76,77].

#### 4.2.4. Induced Pluripotent Stem

Cell reprogramming technology consists of reprogramming adult somatic cells into induced pluripotent stem cells (iPSCs). This technology opened up unprecedented opportunities in the pharmaceutical industry, clinic and laboratories. iPSCs are also expected to be rising stars in regenerative medicine as optimal sources for transplant therapy. iPSCs have similar characteristics to embryonic stem cells (ESCs) in terms of their morphology, self-renewal capacity and differentiation [78], but unlike ESCs, iPSCs are not free from ethical problems. They can also be expanded and used as autologous cells, avoiding the complication of immune rejection [14]; in vitro and in vivo studies on mouse models have demonstrated the enormous potential offered by these cells in generating a number of human autologous cells for advanced chronic wound treatment and degenerative skin disorders. IPSCs can be generated through various methods; Yamanaka and colleagues, in 2006, proposed a method involving somatic cell transduction with a combination of reprogramming factors (e.g., Oct3/4, Sox2, Klf4, and c-Myc,). In 2019, Wang and colleagues highlighted NANOG and LIN28′s potency [78,79,80,81,82]. iPSCs are able to generate any desired cell type, including fibroblasts, keratinocytes and melanocytes; however, reprogramming adult somatic cells and inducing subsequent differentiation in the desired cell line is very difficult. In addition, traces of epigenetic memory involve alterations in genomic stability and differencing capability in iPSC lines [79]. These alterations may cause a highly heterogeneous cell population with undifferentiated iPSCs, their self-renewal results in vivo in teratoma forming. Several efforts were made to derive functional stem cells from iPSCs and MSCs and iPSC–MSC turned out to be an excellent candidate for clinical uses [80]. Exosomes derived from human iPSC–MSC were tested in a rat model and they helped cutaneous wound healing through paracrine signaling, resulting in accelerated re-epithelialization, reduced scar widths, and the promotion of collagen maturity [81]. Reconstituted hair follicle epithelial components of the interfollicular epidermis were observed using epithelial stem cells derived from iPSC (iPSC–EpSC) in the skin of immune-deficient mice [83]. These iPSC-based approaches can produce a great number of human autologous cells; moreover, they could be employed in genome editing techniques. A permanent corrective therapy for chronic injuries resulting from genetic predisposition can be employed by using correct autologous iPSCs. Genome-editing tools can be used for repairing different genetic mutations in iPSCs, such as zinc nuclease finger (ZFN), transcription activator-like nuclease effector (TALEN), or clustered, regularly interspaced, short palindromic repeats (CRISPR). In any case, even if in vitro and animal model iPSCs have led to valid results, these cannot be directly translated into clinical approaches since a greater understanding of these cell types is still needed to protect patient safety.

#### 4.2.5. Skin Tissue Engineering

All these types of approaches adopted by regenerative medicine can be enclosed in the technologies offered by tissue engineering. The term tissue engineering has numerous definitions, but a broad one that we prefer is “*methods that either promote biologic regeneration or repair of tissues, by providing signaling, structural, or replace tissue function with systems that contain living tissue or cells*”. The main elements of tissue engineering are biomaterials, cells, growth factors, other signaling molecules, and engineering components such as scaffolds, pumps, tubes, bioreactors, and oxygenators [84]. Recent technologies in the multidisciplinary field of tissue engineering exploit 3D scaffolds as a key component in the wound healing process. According to the definition of tissue engineering described in a National Science Foundation seminar, scaffolds are the best materials to restore, maintain and improve tissue function [85]. These matrices play a unique role in repairing and, especially in tissue regeneration, providing a suitable platform for various factors associated with cell survival, proliferation and differentiation [86]. Scaffolds can be made from natural/synthetic biomaterials, either materials that remain stable in a biological environment or materials that degrade in the human body [87]. Several techniques have been used for their construction, but the four main approaches used include: (i) sheets of cells secreting ECM; (ii) pre-made porous scaffolds of synthetic, natural and biodegradable biomaterial; (iii) decellularized ECM scaffolds and (iv) cells entrapped in hydrogels, as shown in Figure 4 (schematic representation of classical tissue engineering approach) [86]. In the following paragraphs, different types of scaffolds used in the field of skin tissue engineering, as well as techniques for their design and biomaterials, are described in detail. In addition, their advantages/disadvantages and, above all, their future prospects in the field of skin tissue regeneration are analyzed.

## 5. Scaffold for Skin Regeneration

### 5.1. Scaffold Property and Relationship with Skin Environment

The main objective of the scaffold is to represent a matrix as similar as possible to native ECM; in fact, isolated cells are hardly able to spontaneously organize themselves into new tissues. All cells are in close contact with ECM, either continuously or only at important stages of their development. ECM is well known for its ability to provide structural support to cells and tissues stimulating migration, proliferation, apoptosis, survival and differentiation [88]. These processes are complex and need to be tightly regulated to maintain tissue homeostasis, especially in response to injury [89]. This underlines the complexity in the realization of a scaffold and all the studies necessary to achieve a structure similar to physiological ECMs. Nowadays, numerous approaches are used for designing matrices, consisting of increasingly innovative biomaterials. Until recently, the only two characteristics that a biomaterial had to possess were biocompatibility (the material must be neither cytotoxic nor immunogenic) and biodegradability (the material must be easily eliminated once its function is fulfilled). However, in the modern sense of biomaterials, we must also add the ability to interface with a biological environment and specifically modulate cellular response. The biomaterial becomes, therefore, not only a support for tissue regeneration or a platform for drug delivery, but an active part of cellular function regulation. Based on these assumptions, different parameters have to be taken into consideration such as the physico-chemical properties of pristine materials, mechanical properties, scaffold shape, structure, pore sizes and their distribution.

#### 5.1.1. Structural Characteristics

Skin architectural and mechanical complexity and its properties depend on specific anatomical regions, making scaffold design and production challenging. [90]. Scaffold physico-chemical characterization involves morphology, porosity, water contact angle, mechanical properties, chemical bonds, stability upon incubation in simulated physiological fluids and cell culture medium, while in vitro biological characterization addresses the testing adhesion, adhesion and migration of human cells. To assess these aspects, indirect analyses of immunofluorescence, confocal laser scanning microscopy (CLSM), environmental scanning electron microscopy (ESEM), transmission electron microscopy (TEM) and rheological measurements are conducted. Matrix topography significantly influences cell behavior: the substratum topographical structure has evident effects on the ability of some cellular elements to orient, migrate and reorganize their cytoskeleton. Fluorescence recovery after photo-bleaching technique (FRAP) is exploited to determine material transport as solutes and nutrients. Manufacturing techniques for producing synthetic substrates with controlled surface topography are numerous and quite varied (photolithography, polymerization within a molded preform, physical or chemical deposition from vapor phase, etc.) [91].

Topographical elements on the scaffold surface may have different shapes, sizes, symmetry and/or regularity; these features should be compatible with the material itself and the manufacturing technique. Any kind of depression or surface relief, even porosity and roughness, define the topographical structure of scaffold surface [92]. Channel structure is the topographic typology most used for experiments with living cells. In general, experimentation with surfaces on which channels have been produced has shown alignment of cells with the main axis of channels and often the organization of cytoskeleton components and focal contacts oriented in the same direction [93]. Finally, the symmetry and regularity of the topographic structure, together with the other morphological and dimensional characteristics, are important properties for cells. Scaffolds must have sufficient mechanical resistance and optimal elasticity both during in vitro culture, to maintain the spaces required for cell growth, and during in vivo application, to fit with damaged tissue. Currently, the mechanical properties of engineered skin are as close as possible to human skin, but they still do not fully correspond to native skin features. The greatest disparity is observed in vitro and in early stages of grafting, in which engineered skin turns out to be weaker than native skin [94]. This mechanical deficiency causes a difficulty in surgical application, decreasing the elasticity and resistance after grafting. Matrix structural properties, during the degradation process, should also be considered and appropriately adjusted in order to maintain an adequate structural integrity for a sufficient time, so that the newly developed tissue can replace its supporting function. Scaffold porosity should be compatible with mechanical properties and promote the cell growth coordination. Moreover, a scaffold with pores of adequate sizes improves cell migration, water absorption and endorses oxygen mass transfer into the scaffold [95]. The traditional tissue engineering approach exploited rigid scaffolds (top down approach), but thanks to 3D bioprinting, the efforts of the scientific community are moving towards a bottom-up approach. This procedure involves the elimination or at least the reduced use (scaffold-less) of traditional scaffolds. One innovative feature of bioprinting involves a temporary scaffold.

Different parameters, such as wettability and surface features, influence the biological performance of polymer-based biomaterials. Wettability is generally estimated by measuring the contact angle; it describes aqueous fluid’s capability to spread on the scaffold surface and into its pores, it is directly related to intermolecular forces between phases and it guarantees cell–cell and cell–substrate interactions. Cell adhesion, migration and proliferation are generally improved by controlling wettability and scaffold surface features. These last features may be modified by different techniques, including surface topography modification and chemical treatments [96].

#### 5.1.2. Biological Characteristics

Biological aspects include biocompatibility, nontoxicity and biofunctionality. Scaffolds allow tissue formation without causing interference [97]. Several efforts have been made to realize not only biocompatible and biodegradable scaffolds, but also bioactive matrices to promote proper cellular interaction, migration and differentiation. Biocompatibility is the discriminating factor between a biomaterial and a material; the biomaterials improve and/or restore certain biological functions without interfering or interacting in a harmful way with physiological activities. The materials can be divided into permanent and biodegradable materials. The first ones resist biological environment action, preserving their features; biodegradable materials undergo a progressive mass loss or degradation after implantation in the human body. The degradation byproducts can produce inflammatory reactions, which is a possible disadvantage of biodegradable materials. Bioabsorbable or bioresorbable materials, on the other hand, are those materials that undergo progressive degradation; however, the degradation products are compatible with the human body and/or its metabolic pathways. The degradation time should be closely synchronized with that of tissue growth and its formation, as too rapid reabsorption does not allow the formation of a complete and robust tissue; too long scaffold degradation times, on the contrary, induce the formation of fibrotic tissue around the scaffold. A further characteristic of biomaterials is their bioactivity, namely the ability of the material to induce a specific biological activity in the organism. This type of material allows the formation of biochemical bonds and direct interactions with biological tissue growing freely on the surface [98].

### 5.2. Biomaterials for 3D Scaffolds

#### 5.2.1. Synthetic Biomaterials

Synthetic materials have two significant advantages: they can be reproduced industrially on a large scale, controlling parameters such as molecular weight and degradation time; and their chemical properties can be modified to produce derivatives with improved adhesion, cross-linking and biodegradability properties [97]. Polymeric materials, in particular biodegradable polyesters, thanks to their greater compatibility with body tissues, are the most used in tissue engineering. Synthetic polymers, such as polycaprolactone (PCL), polylactic acid (PLA), polyglycolic acid (PGA), and related copolymers (polylactic-co-glycolic acids (PLGA)) can compose matrices individually or as composites. PCL was initially used for the production of degradable films and molds, and today it is widely used in various sectors such as in the production of absorbable sutures, drug delivery systems and scaffolds for tissue regeneration. Linear aliphatic polyester undergoes hydrolytic degradation in a physiological environment due to its interaction with water molecules. Since its glass transition temperature (Tg) is ~−60 °C, it is in the rubbery state at room temperature and after implantation and its high plasticity can be modulated by blending with other polymers such as PLA, PGA and PLGA in order to increase its stiffness. PGA is a crystalline polymer and therefore is not soluble in many organic solvents. Due to its hydrophilic nature, PGA has a tendency to rapidly lose its mechanical strength and it is re-absorbed in about 4 weeks after implantation. PLA, although structurally very similar to PGA, differs from the latter in its chemical, physical and mechanical properties, due to the presence of a methyl group at the α carbon that provides a higher hydrophobicity and reduces its hydrolysis rate [86].

Composite systems containing synthetic polymers and bioactive substance were studied for improving cell growth and healing effectiveness [99]. Bioactive glasses (45S5 Bioglass^®^) were blended with the PLGA copolymer and showed encouraging evidence in in vitro and in vivo scaffold neovascularization. Poly (3-hydroxyoctanoate) nano-sized 45S5 Bioglass^®^ scaffolds, characterized by their high wettability and rough surface, provide a microenvironment suitable for cell growth and accelerate blood clot time. Mesoporous bioactive glass (MBG) electrospun fibers based on poly(ethylene oxide) were researched for supporting skin tissue regeneration and controlling anti-inflammatory agents [86].

#### 5.2.2. Natural Biomaterials

Natural materials have been recently studied with the main purpose of overcoming the limitations of synthetic materials. They have the advantage of containing, in their structure, signal sequences promoting and maintaining cell adhesion and functions. Collagen, gelatin, silk fibroin and fibrin are the most used natural biomaterials of protein derivation and are intended for tissue engineering. These polymers are already present in the human body as ECM-forming elements. The most common technique consists of extracting the materials from human or animal sources; however, they are not available in large quantities and may be a vehicle for pathogens. For these reasons, advances in biotechnology have made possible their production through the fermentation of microorganisms and in vitro enzymatic synthesis.

Being of biological origin, collagen has good mechanical properties and biocompatibility, it is susceptible to crosslinking and its degradation occurs physiologically. Unfortunately, biological characteristics also include disadvantages, such as rapid degradation by collagenase, gelatinase and other proteins. Moreover, it is susceptible to any sterilization process [86]. Collagen hydrogels, microfiber collagen scaffolds and electrospun collagen nanofibrous scaffolds are just a few examples of formulations for skin regeneration. Collagen has been exploited to produce nanofibrous scaffolds, by electrospinning, as skin substitutes of collagen type I and type III, or to coat scaffolds made from other materials and increase their biocompatibility [86,100]. Gelatin is a partially hydrolyzed version of collagen wherein the triple-helical structure of collagen is changed into single-stranded molecules. It contains a high number of amino acids such as glycine, proline and 4-hydroxyproline. Gelatin is advantageous compared to collagen because of its lower immunogenicity and greater cellular adhesion potential, due to the presence of arginine–glycine–aspartic acid sequences [101]. Scaffolds consisting of gelatin nanofibers showed possible applications in wound healing regeneration. Several gelatin formulations such as gelatin–alginate sponges, gelatin containing EGF and gelatin films have shown potential applications in the treatment of burned skin tissue [86]. From a mechanical point of view, collagen and gelatin do not have a high mechanical resistance and, consequently, they cannot be used as scaffold for hard tissue.

An innovative and increasingly used biological polymer in tissue engineering is silk fibroin, a fibrous protein-forming silk core. Fibroin is a protein made from β-antiparallel leaflets rich in alanine and glycine. It is an interesting biomaterial for tissue engineering and it can be exploited to produce scaffolds in the form of fibers, hydrogels and sponges [86].

Fibrin is another promising material in the field of bioengineering. It is obtained from fibrinogen directly extracted from patient, it is immunocompatible and it is used as a temporary scaffold [102]. Fibrin gels are used for the release of growth factors, cytokines or other bioactive molecules that control adhesion, proliferation, cell migration, differentiation, and extracellular matrix production. Many growth factors are able to bind with fibrin, especially BFGF and VEGF [103].

Biomaterials such as keratin, bovine serum albumin, eggshell and membrane proteins, have been proposed for producing skin regeneration products. Keratin and its derivatives have been employed for producing scaffolds as platforms for antibiotic or growth factor delivery, while eggshell provides an ideal ECM environment for human skin fibroblast cells [104].

Homoglycan and heteroglycan polysaccharides as were widely investigated for skin regeneration; in particular, D-glucans were used for producing soft scaffolds able to support effective wound healing and limited skin irritation [105]. Chitosan (CS) promotes collagen and hyaluronic acid synthesis, and deposition into the wound site. It also stimulates fibroblast proliferation, cytokine production through macrophage activation and angiogenesis; in addition, CS has been shown to have microbicidal activity against bacteria [86,106]. Alginate is a water-soluble heteroglycan polysaccharide characterized by its high water absorbing ability; it is exploited for maintaining a suitable moist environment at the site of injury.

Hyaluronic acid (HA) is essential for skin regeneration. HA hydrogel scaffolds are well known for directing tissue regeneration by supporting angiogenesis and neuritis outgrowth/repair. The polymer has a high viscosity and surface tension, parameters that can represent a challenge in designing HA scaffolds [86].

Cellulose is a nanostructured biomaterial; its similarity to ECM has attracted great potential in wound healing scaffolds for several applications in tissue repair, remodeling or skin transplantation [107].

Carbon-based nanomaterials were also applied for tissue engineering; their biocompatibility and incredible mechanical strength make them promising candidates for skin treatments. Carbon-based scaffolds are compatible with natural ECM, promoting cell–cell interaction and normal cellular functions in tissue engineering. Graphene is an example of a carbon-based nanomaterial, it is an allotrope of carbon and its structure consists of a single carbon atom planar sheet on which each atom is bound to three others, inside a crystalline honeycomb lattice. Graphene has exceptional physico-chemical and mechanical properties, specifically its high Young’s modulus value (~1.0 TPa), large surface area (2630 m^2^·g^−1^), large intrinsic mobility (200,000 cm^2^·v^−1^·s^−1^), good conductivity temperature (~5000 W·m^−1^·K^−1^), high optical transmittance (~97.7%) and excellent electrical conductivity [108].

Natural biomaterials can be used as they are or in combination with other biological or synthetic materials and they can be combined with bioactive molecules or drugs. An example of a complex matrix is reported in the study conducted by Singaravelu and colleagues, where a porous keratin–fibrin–gelatin 3D sponge scaffold (KFG-SPG), was designed and the drug mupirocin (D) was incorporated to realize a KFG-SPG-D scaffold for tissue engineering applications [8].

## 6. Scaffolds for Skin Tissue Engineering and Cutting-Edge Techniques for their Manufacturing

Several technologies such as phase separation, solvent casting and particulate leaching, membrane lamination, melt molding and high-pressure processes have been studied and proposed for producing 3D scaffolds. Below, those recognized as the most innovative ones are described.

Scaffold-free skin equivalents are a promising approach for exploiting cellular functions and their secreted ECMs. Scaffold-free skin equivalents can be produced by cell sheet technology, which is one of the most advanced methodologies due to its process simplicity and versatility; moreover, the final product is characterized by excellent compatibility with native skin tissue, and poor foreign body rejection. The scaffold-free skin equivalents address several technical challenges such as prolonged in vitro culture incubation and limited implantation volume, as well as intrinsic physical weakness and poor vascularity [14]. Therefore, biomaterials and tissue engineering technologies have been introduced to overcome these issues.

### 6.1. Cell-Free Scaffold

Cell-free scaffolds are fabricated by decellularization techniques using either chemical, physical or enzymatic degradation, including repeated freeze–thaw cycles, hypertonic or hypotonic and trypsin/EDTA treatment, etc. [14]. The human dermis decellularization procedure allows us to achieve good biocompatibility and less immunogenicity in dermal replacement; nevertheless, it is well demonstrated that the decellularization protocol negatively influences the matrix structure and orientation. High concentrations of detergents and enzymes are employed and, following washing steps, are required to remove the residuals. Milan and colleagues showed an innovative approach to address the issue related to standard decellularized protocols. They used an optimized surfactant (n-octyl-beta-D-glucopyranoside) at low concentrations as a tissue pretreatment to avoid enzyme digestion and preserve ECM. This optimized and facilitated cell membrane structure removal and lipid and protein solubilization [109].

Decellularized animal tissues represent a valid alternative for overcoming the limits of human donor availability. Jafarkhani and colleagues obtained a decellularized scaffold from a bovine heart; the cell-free prototype was also functionalized with graphene oxide (GO) and engineered with skin cell lines (NIH/swiss mouse embryo fibroblast cells, NIH 3T3). The in vitro results proved the capability of the prototype scaffold to form a microenvironment for cell proliferation, differentiation, migration and gene expression and the authors demonstrated its applicability and suitability as a support material for tissue engineering [110].

### 6.2. Electrospun Scaffolds

Electrospinning is a cutting-edge technology that allows us to effectively produce ultra-thin fibers with diameters ranging from submicrons to a few nanometers. Briefly, an electrically charged polymer solution is forced to pass through a nozzle and is influenced by an electric field. When a sufficient voltage is achieved between the nozzle and metallic collector, the liquid droplet at the nozzle becomes charged and opportunely stretched in a continuous jet. The jet travels to the collector, and it dries out in flight [111]. Electrospun scaffolds resemble native ECM design, variable pore size, high surface area and oxygen permeability, making them suitable as skin replacement materials. Moreover, nanofibers can be loaded with bioactive substances, namely growth factors, nanoparticles, antimicrobials, anti-inflammatory agents and wound healing drugs. Several natural and synthetic polymers, as well as blends containing them, have been studied for producing electrospun nanofibrous membranes, such as gelatin and polycaprolactone blending [112]. Nanotechnology can also be exploited for creating innovative drug delivery systems, designed to achieve a controlled release of an active agent. Electrospun nanofibers showed promising results for the local release of antibiotics, preventing bacterial biofilm formation and limiting antibiotic resistance. Antibiotic-loaded electrospun matrices could be applied in several applications: severe burns, gingival cavities for local infection treatment or for preventing post tooth explant infection, to name a few. Antibiotic-loaded electrospun matrices allow us to reach high antibiotic concentrations at the site of infection, avoiding high system concentrations and related side effects. Moreover, the administration frequency is reduced and, consequently, patient compliance enhanced [113].

The antibacterial activities of metal oxide nanoparticles are well known and they can be loaded in electrospun matrices for incremental wound healing. Metal oxide nanoparticles inhibit bacterial growth, producing reactive oxygen species (ROS) that cause detrimental oxidative stress. ROS concentration can play a key role in cell proliferation and migration, apoptosis, and wound healing. Several metals (silver, gold, etc.) and metal oxide nanoparticles (titanium and zinc oxide) have been extensively studied for their wound healing and microbicidal properties.

The bioactivity of PCL fibers was improved by incorporating metal hydroxide nanoparticles such as europium hydroxide (EHN); the nanoparticulate system enhanced both endothelial cell density and cell growth. The blending of metal hydroxide nanoparticles and PCL polymers is a promising approach for promoting cell proliferation as well as blood vessel formation and wound healing rate. It was also demonstrated that nanoparticles based on silver (AgNPs) inhibit the growth of *Staphylococcus epidermidis* and *Staphylococcus haemolyticus* [114], and electrospun membranes containing silver nanoparticles exhibit microbicidal activity against Gram-positive and Gram-negative bacteria strains. Moreover, gold nanoparticles, due to their intrinsic antioxidant properties, were beneficial for cutaneous wound repair [115]. PCL/gelatin nanofibrous membranes containing 6-aminopenicillanic acid (APA) and gold nanoparticles (Au-APA) showed good healing properties and antibacterial activity against multidrug-resistant bacteria [116].

### 6.3. Three-Dimensional Bioprinting

Traditional scaffolds are substrates where cells can adhere to proliferate, and their porous conformation is an ideal cell deposition environment. In spite of these benefits, the use of traditional scaffolds has some limitations: the vascularization of engineered tissues; correct and precise cell deposition in the porous structure of the scaffold; and the stiffness of the support matrix. Moreover, an inflammatory process occurring during the degradation of a biodegradable scaffold can sometimes limit the new tissue growth rate. Overall, it is practically impossible, via traditional techniques, to obtain tissues formed by layers of different cell densities that simulate the complexity of a multi-tissue structure [117]. It is therefore clear that these limitations are intrinsic in the use of traditional scaffolds as supports for tissue growth. Three-dimensional bioprinting is an emerging technique for engineering biological constructs that involves dispensing cells into a biocompatible matrix (bioink) using sequential layer-by-layer computer-aided design (CAD) and generates a tissue-like 3D structure. The technique can overcome several drawbacks of traditional scaffold manufacturing techniques, but its immediate novelty is its ability to generate an engineered tissue in a single-step process [118]. Three-dimensional bioprinting processes include magnetic bioprinting, stereolithography, photolithography, and direct cell extrusion. Briefly, cells are retrieved from the patient and proliferate in vivo. Before seeding, cells are phenotypically characterized and suspended in culture media and in a biomaterial solution to get a suitable bioink. The bioink is then printed based on a medical scan of the patient (Figure 5). The engineered construct is incubated in a bioreactor, ensuring proper nutrients and gas flow for the maturation of the engineered construct into functional tissue [119].

Some of the most important and characteristic aspects of 3D bioprinting area as follows. Drop-on-demand: this consists of the possibility to deposit drops of bioink through a computerized control system, only when and where necessary to recreate an image or a desired sequence. It also allows the control of density and the very high customization of the cell solution deposition. The resolution of the final product depends on the print head, the minimum size of the droplets and their capability on the surface [119].

Solid freeform fabrication: bioprinting has no limitations on the shape and consistency of the desired structure. By the automatic processing of computer-aided design (CAD) images and computer-aided manufacturing (CAM), it is possible to realize complex 3D structures from data and images from medical investigations, such as magnetic resonance imaging (MRI) and computed tomography (CT). Such images can be processed with technologies such as Rapid Prototyping (RP) or Solid Freeform Fabrication (SFF) that allow the computer to predefine the microscopic and macroscopic shape of the scaffold [120]. Layer-by-layer: bioink drops are deposited on the first biopaper layer, after which another biopaper layer is superimposed, which, in turn, is seeded with cells. By iterating this process, a 3D structure is formed, consisting of a 2D sheets of biopaper, which, by merging together, will form a single system [119].

The production of bioink is divided into two steps: cell culture and the preparation of bioink itself. Moreover, another innovative feature of 3D bioprinting is the newly developed, temporary and bioabsorbable (biopaper) scaffold. Biopaper is a support surface where cells can be seeded until they have fused, multiplied and organized into a real tissue. In addition, biopaper must encourage a continuous and constant supply of nutrients and oxygen to cells. From a structural point of view, it is usually composed of hydrogels that simulate the cellular environment; the hydrogel material can be from synthetic or natural polymers. Biopaper is presented in liquid form but, after being laid, it changes to a solid–gelatinous consistency [86]. There are many hydrogel compositions that are used in 3D bioprinting to incorporate and provide cells for skin repair; those of natural origin use mainly alginate, collagen type I, HA and chitosan, while those of synthetic origin employ polyvinyl alcohol PVA), polyethylene glycol (PEG), polycaprolactone (PCL), polylactic acid polymers (PLA) and their compounds [14]. Three-dimensional bioprinting can be used to generate multi-layer vascularized human skin grafts that can potentially exceed the survival limits of the grafts observed in current avascular skin replacements. In a study conducted by Baltazar and colleagues, it was shown that 3D bioprinting succeeded in fabricating vascularized human skin in vitro from human cells, with morphological and biological similarity to in vivo human skin. An implantable skin graft, including a perfusable microvascular system, was obtained by 3D printing; the prototype was made by combining a bioink with foreskin dermal fibroblasts (FBs) and human endothelial cells (ECs), with a second bioink including foreskin keratinocytes (KCs) to form the dermis and epidermis, respectively [121]. Won and colleagues conducted another innovative study on decellularized ECM derived from a pig dermis; the authors obtained a printable bio-ink that was mixed with human dermal fibroblasts (HDFs) to produce a construct loaded with human cells. The residual ECM containing collagen and GAG, as well as bioactive molecules and growth factors, provided cells with an environment identical to that in the tissue, helping with the viability and proliferation of HDF in the construct [122].

#### Emergence of 3D Bioprinting for Skin Regeneration

Three-dimensional bioprinting is a promising technology for skin regeneration; it is applied to produce bio-artificial skin by the stratification of autologous cells previously isolated and proliferated in a laboratory. The stratification can be achieved by using a 3D scaffold, or cell spheroids. This technology allows us to control the geometry at the micro/nano-cellular level, which is essential for modulating cell–substrate and cell–cell interactions. This methodology could be particularly valuable for skin tissue engineering, as it allows to accurately lying out a multilayer 3D scaffold engineered by layer-by-layer technology, assembling different cells.

The biomimetic technique is defined as a biological-inspired technique and it is employed to imitate different functional cellular elements, such as multilayer skin and vascular system branching [86]. A thorough knowledge and the collaboration of imaging, biomaterials, biophysics, engineering, medicine, and cell biology are required to achieve a significant breakthrough using this approach.

A knowledge of organogenesis, together with ability to manipulate the microenvironment to force cell differentiation and bioprinted tissue production, have allowed the development of cutaneous tissue through autonomous self-assembly, a technique able to reproduce a biological tissue following the map of embryonic development. This technique involves the 3D bioprinting of cell spheroids, which undergo cell fusion and cell reorganization, to mimic the architecture of the developing tissue. Spheroids can vary in size, depending on the parameters set by the user. Their complete biological function depends directly on the cells that secrete their ECM component, following the signaling pathways for histogenesis, and on the localization process.

Mini tissue has been proposed as an alternative self-assembling approach. In this context, 3D bioprinting is exploited to form macro-constructs, assembling mini tissue based on cellular spheroids [123].

In general, tissue engineering shows infinite potentialities, but several constraints still need to be addressed. The design of a complex hierarchical structure engineered with different cells is ongoing and challenging; a few technologies have emerged, such as assisted laser bioprinting (LaBP) and laser-induced transfer (LIFT), for producing 2D and 3D constructs incorporating different cell lines. LIFT technology was used for assembling fibroblasts/keratinocytes and MSCs in a single 3D construct; the impact of the production phase was estimated by quantifying the cell survival rate, cell surface marker changes and DNA damage. The data demonstrated that fibroblasts, keratinocytes and human mesenchymal stem cells (hMSCs) were able to survive during the production phase, and they retained their proliferation ability with no evidence of DNA or surface marker alterations [86].

### 6.4. Bioreactors

Bioreactors are devices where biological and/or biochemical processes take place under carefully and strictly controlled operating and environmental conditions (pH, temperature, pressure, feeding, and removal of waste products). In these systems, cell-engineered scaffolds complete their cellularization before being implanted. Compared to the progress that has been made in the design of scaffolds, much progress has also been made in bioreactors, especially for creating devices capable of overcoming limitations to nutrients and oxygen transport that hinder the achievement of in vitro engineered tissues suitable for clinical applications. Bioreactors are used either in the engineered tissue maturation phase or during cell seeding, in order to overcome static seeding limits that preclude a uniform cell distribution along the entire scaffold thickness. The bioreactor provides an important step towards the achievement of functional grafts; it efficiently supports cell nourishment and, if combined with mechanical stimuli application, it directs cell activity, cell functions and differentiation. In addition, bioreactors offer well-defined culture conditions that are useful for systematic and controlled cell differentiation and tissue development studies (Figure 4). Computational modeling and experimental tests were used to study transport phenomena within the bioreactor [124]. In a study conducted by Navarro and colleagues, the authors developed a dual-chambered membrane bioreactor for the co-culture of stratified cell populations (DCB) to study 3D-stratified cell populations for skin tissue engineering. DCB provides adjacent flow lines within the chamber and the included membrane regulates stratification and flow mixing. This system can be exploited to produce cell population layers or gradients in scaffolds [125].

## 7. Discussion

The collected data highlight the complexity of the wound healing process depending on the type of wound, its persistence (chronic wound, acute wound) and cause. Wounds have a variety of causes; some result from surgery and injuries, and others are a consequence of extrinsic factors (e.g., pressure, burns and cuts), or pathologic conditions such as diabetes or vascular diseases. These types of damage are classified into acute or chronic wounds depending on their consequences of their underlying causes. Complete and timely wound closure is the main objective of all aspects of wound care, although this is not always achievable. Chronic wounds, in particular, are challenging for effective wound care.

A distinction in wound healing treatments was made in this review between wound care and skin regeneration.

Recent knowledge about wound care has highlighted advanced dressings, which, along with other techniques and technologies, are able to help in the management of chronic wounds. Advanced medications play an active role in treating serious injuries having complex healing profile. Regardless of the nature of the lesion and the medications used, an ideal wound dressing should provide biological compatibility, water adsorption and retention properties, low cytotoxicity, a nonstick ability, conformability and antibacterial effects. It should prevent the wound from being infected, allow gas exchange, adsorb excrescent wound exudates, and retain a part of the exudate to maintain the local moisture of the wound, accelerating the healing.

Skin regeneration refers to more reconstructive treatments that aim to restore the complex tissue and are more suitable for chronic injuries that have not obtained benefits from other less invasive strategies, such as the application of advanced dressings or therapies that use physical/mechanical energies. Skin regeneration products represent the most innovative branch in skin injury treatments and involve a multidisciplinary approach.

Scaffolds represent an innovative strategy among the various innovative strategies for skin regeneration and the delivery of drugs to promote wound healing. A scaffold combines the concepts of cell therapy and pharmacology with tissue engineering. Scaffolds activated with growth factors, bioactive molecules and genetically modified cells could overcome the limitations of current wound healing technologies and design personalized therapies for patients. Emergent technologies for scaffold manufacturing include electrospinning and 3D bioprinting. The electrospinning process represents a cutting-edge technology for producing ultra-thin fibers with diameters ranging from submicrons to a few nanometers. The structure of electrospun sheets resembles the native ECM architecture, with a controlled pore size, high surface area and suitable gas permeability; moreover, bioactive substances (such as growth factors, antimicrobials, anti-inflammatory and antifibrotic agents and wound healing drugs) can be incorporated to improve the wound healing process. The electrospinning process allows the production of either advanced medication or 3D scaffolds for skin regeneration, depending on the electrospun matrix design and composition. It is highly reproducible and scalable at an industrial level; thus, it is a promising technique for future development. Advances in 3D bioprinting can offer precise models for cell growth and allow the recapitulation of native skin architectural organization. The aim is to design bioactive scaffolds engineered with different cells, including fibroblasts, keratinocytes, epidermal progenitors and endothelial cells, for faithfully mimicking the skin microenvironment.

## 8. Conclusions

Despite the enormous advances in wound healing therapies in recent years, the most difficult challenge in skin regeneration is to obtain a tissue containing microvessels, hair follicles and sweat glands. Three-dimensional bioprinting has an important potential for wound healing, but the challenge remains in designing a precise and complex hierarchy in new tissues, allowing different types of cells to coexist and to have unique 3D models. Therefore, one of the key areas for 3D bioprinting improvement is to provide bioinks suitable for different types of cells. Biodegradable and biocompatible polymers of natural origin are the preferred candidates and combinations. Technological limitations due to cell damage caused by the 3D bioprinting process and due to the rheological properties required to obtain a 3D bioprinted construct can be overcome by new 3D bioprinter designs. A great opportunity in the near future lies in combining 3D bioprinting with electrospinning. This will enable the possibility of manufacturing scaffolds with improved mechanical resistance properties and cell engineering.

Given the extent and increasing prevalence of various types of skin injuries, skin regeneration is a challenge that requires close collaboration between researchers in many disciplines. Many innovative strategies have been implemented to address wound healing due to the easy accessibility of target tissue; the design of innovative and more functional strategies to translate technologies into clinical therapies represents a great challenge for biomaterial researchers and biomedical engineers.

Advances in research will lead to improved wound healing treatments in terms of ameliorating skin reconstruction and scar formation. The impact on real-world outcomes will be an improvement of life-threatening situations, with the aim being to restore to normal life conditions, even for patients suffering from severe wounds of different origins.

## 9. Future Prospective

Recent emerging cutting-edge technologies provide innovative pathways and strategies to improve severe wound treatments. The current scenario is constantly updated by several efforts carried out in different fields, including interdisciplinary science, material science engineering, and gene therapies, to name few. The progress in skin repair and regeneration is occurring very quickly as result of novel technological improvements in biomedical research technologies, such as disease pathways, advanced medical device technology and therapies. These innovations can support better science, as well as faster medical treatment development, a reduction in costs and experiments on animals. The innovative techniques recently exploited, in combination with conventional methods, offer a great perspective for the future. In particular, the innovative trend of regenerative medicine, together with personalized medicine, may facilitate advanced wound treatment development, leading to rapid healing, as well as painless and scarless treatment.

## Figures and Tables

**Figure 1 pharmaceutics-12-00735-f001:**
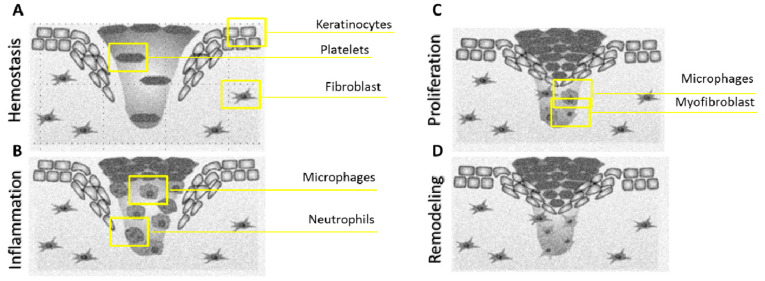
Schematic representation of wound healing process with cells involved in each phase.

**Figure 2 pharmaceutics-12-00735-f002:**
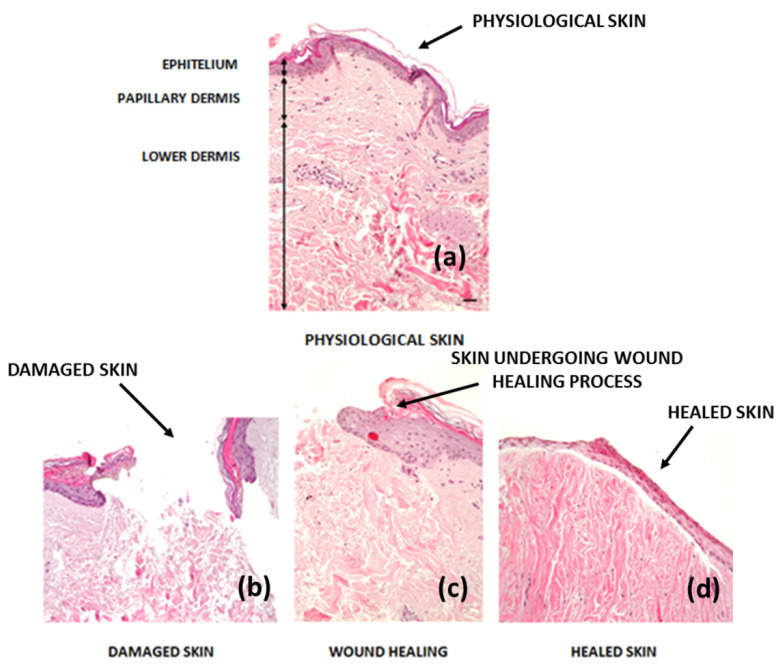
Stages of wound healing process: (**a**) physiological skin, (**b**) damaged skin, (**c**) skin undergoing wound healing process and (**d**) healed skin. Histological image, hematoxylin–eosin, 10×, bar: 20 µm, the red arrows highlight the named stages. Reproduction from: Dpt. Clinical–Surgical, Diagnostic and Pediatric Sciences, University of Pavia.

**Figure 3 pharmaceutics-12-00735-f003:**
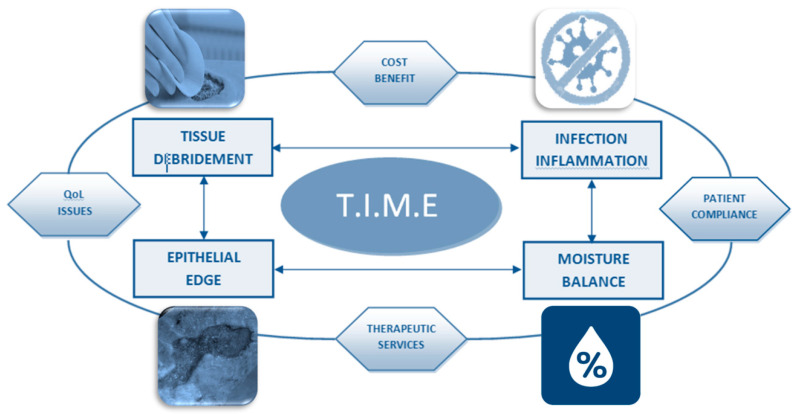
Schematic representation of Tissue (T), Infection (I), Moisture (M) and Epithelial (E) (TIME) concept.

**Figure 4 pharmaceutics-12-00735-f004:**
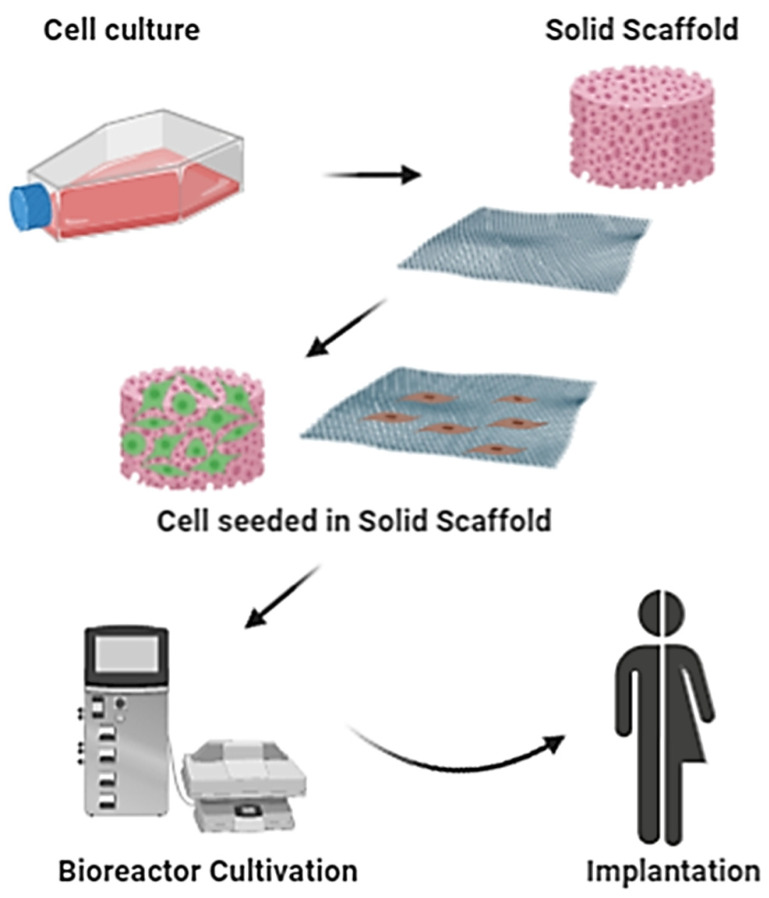
Schematic representation of classical tissue engineering approach.

**Figure 5 pharmaceutics-12-00735-f005:**
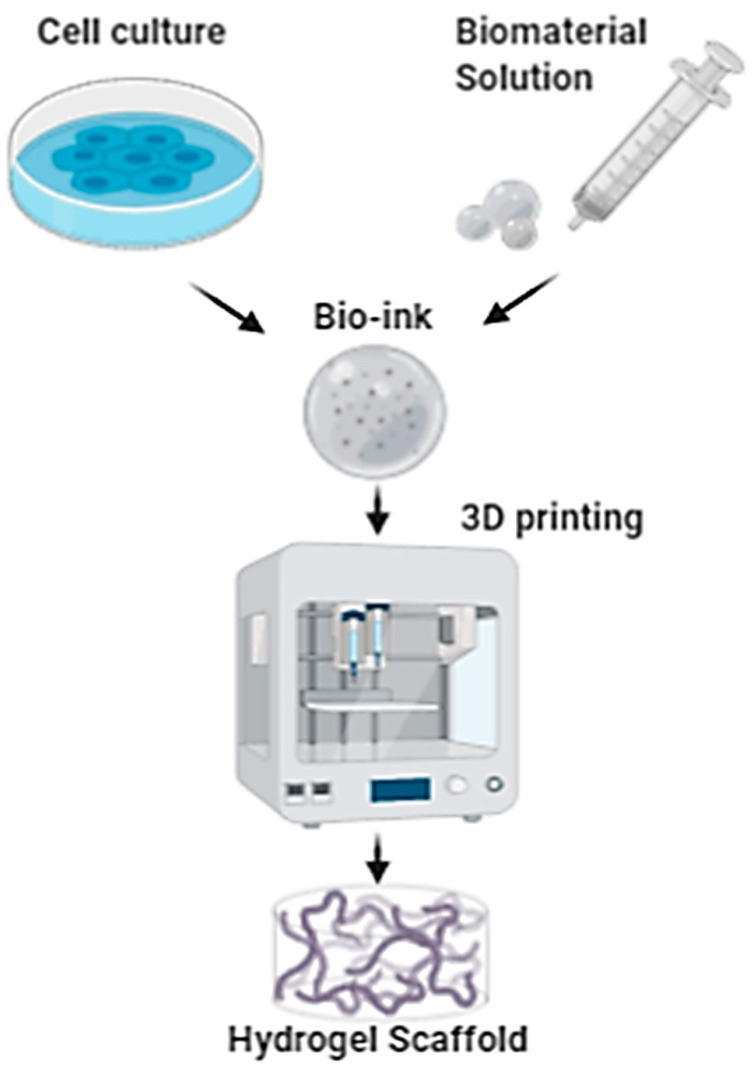
Hydrogel approach. Mixtures of biomaterial solutions, for example, chitosan, cellulose, polyethylene glycol, poly-caprolactone, etc., are mixed with the skin cells to generate hydrogel scaffolds.

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
