# Peer review of "Skin Wound Healing Process and New Emerging Technologies for Skin Wound Care and Regeneration"

_pharmaceutics, 2020, doi:10.3390/pharmaceutics12080735_

Round 1

Reviewer 1 Report

Comments:

  1. Please remove section 2, which does not provide useful information to readers. It is believed that the same information is implicitly reflected in the references.
  2. Authors are advised to create a comparative table on the mentioned “New emergent technologies for skin wound care and regeneration” and emphasize the advantages and disadvantages of each of the emergent technologies discussed in the article, which provide better insight into the current progress of those technologies.
  3. Please give a clear description of the review to improve the main goal, and it should be related to the title and abstract of the manuscript. Some parts seem irrelevant to the main subject of the article.
  4. As a review article, each section lists only a few investigations on the topic, without summarizing the reviewed literature. The literature listed in the article is incomprehensible. For example on Page 1-2, when introducing the wound healing process and the emerging technologies for skin wound care and regeneration, the author only listed few related papers (only 2 references) and did not make a summary of the article title.  Moreover, the authors explained each section elaborately with limited references. Eg., Section 3.2 (3 references). I recommend, refer to the articles in the appropriate section, which will help to strengthen the author’s claims.
  5. The authors give a brief glimpse of chronic wounds and their association with pathogens. In general, major chronic wounds are associated with pathogens and there are well-known reports are available in the literature, I recommend to use in this article under the section 3.3.
  6. Figure 1, The author mentioned “Stages of the wound healing process” and in the respective description in line numbers 274 the author mentioned that “Scar formation ends in the remodeling phase”. In order to be more intuitive to readers, the authors should add an arrow in the image and indicate exactly what is going on in each image.
  7. Kindly see the typo in the figure 1 legend. It must be “Histologic Image“ not “Hystologic image”.
  8. Figure 2: It schematically represents the classical approach of tissue engineering, but the author indicates this figure in the sentences (line number 763-766) are not suitable. Please check and cite Figure 2 appropriately.
  9. Figures 1, 2, and 3: The present form is not clear enough to publish, better resolution images should be provided.
  10. Since the major focus of this review is on “New emergent technologies for skin wound care and regeneration”, authors are advised to incorporate a dedicated section to explain the potential role and the status of skin wound care products (which are produced from the discussed emergent technologies in the review) in clinical and pre-clinical studies and rewrite the discussion accordingly.
  11. I recommend adding one more figure which schematically explains the types of “New emergent technologies for skin wound care and regeneration”.
  12. Write a separate section on “Future perspectives” must discuss here on the recent development in new emergent technologies for skin wound care and regeneration and outlook of the prospects use in tissue engineering and wound healing applications
  13. Please check the Section and sub-section number carefully and correct it, the numbers are repeated in Section 4 and 5. Section 4 previously discusses on “chronic wounds” and the same number given to the manuscript overall “discussion” (see line number 1220), similarly, Section 5 discussed on “skin regeneration process” and the same number given to the manuscript overall “conclusion” (see line number 1264).
  14. I would suggest the authors re-organize their manuscript. The information given in Section 5-7 is more appropriate to the title and abstract of the review article. In the present form, the information given in Section 3 & 4 are very elaborate, although I may ask the author to re-write briefly with providing the necessary outcomes from Section 3 and 4.

Author Response

Pharmaceutics

Manuscript Title: Skin Wound Healing Process and New Emergent Technologies for Skin Wound Care and Regeneration

Authors: Erika Tottoli, Rossella Dorati *, Ida Genta , Enrica Chiesa , Silvia Pisani, Bice Conti

Ms. Ref. No.: pharmaceutics-858731

We are grateful to the Reviewers for their constructive comments and suggestions to improve the manuscript. Our point-by-point responses to the reviewers are presented below:

Reviewer #1

  1. Please remove section 2, which does not provide useful information to readers. It is believed that the same information is implicitly reflected in the references.

Section 2 has been removed as suggested by the Reviewer and number of paragraphs corrected accordingly.

  1. Authors are advised to create a comparative table on the mentioned “New emergent technologies for skin wound care and regeneration” and emphasize the advantages and disadvantages of each of the emergent technologies discussed in the article, which provide better insight into the current progress of those technologies.

The table 1 has been created emphasizing advantages and disadvantages of each emergent technologies discussed in the article. The table has been included (Paragraph 3.2. - Advanced Dressings: the transition of advanced dressings from passive to active role in wound healing process) to facilitate the reading of following paragraphs.

  1. Please give a clear description of the review to improve the main goal, and it should be related to the title and abstract of the manuscript. Some parts seem irrelevant to the main subject of the article.

A clear description of main goal has been included in the manuscript (Paragraph 1 – Introduction) taking in consideration title and abstract. The authors agree with the review, some parts seem irrelevant nevertheless the review has been organized with a multidisciplinary approach involving some secondary aspects like biomaterials and 3D scaffold production, rather than bioreactors. The complexity of new Emergent Technologies in Skin Wound Care and Regeneration requires an integrating multidisciplinary knowledge involving several researchers, to name a few clinicians, biologist, engineers and formulators.

  1. As a review article, each section lists only a few investigations on the topic, without summarizing the reviewed literature. The literature listed in the article is incomprehensible. For example on Page 1-2, when introducing the wound healing process and the emerging technologies for skin wound care and regeneration, the author only listed few related papers (only 2 references) and did not make a summary of the article title. Moreover, the authors explained each section elaborately with limited references. Eg., Section 3.2 (3 references). I recommend, refer to the articles in the appropriate section, which will help to strengthen the author’s claims.

As suggested by the Reviewer references have been equally distributed in all manuscript.

  1. The authors give a brief glimpse of chronic wounds and their association with pathogens. In general, major chronic wounds are associated with pathogens and there are well-known reports are available in the literature, I recommend to use in this article under the section 3.3.

In revised version, section 3.3 (Acute and chronic wound healing) has been numbered as 2.3. As suggested by the Reviewer further references have been included in the manuscript.

  1. Figure 1, The author mentioned “Stages of the wound healing process” and in the respective description in line numbers 274 the author mentioned that “Scar formation ends in the remodeling phase”. In order to be more intuitive to readers, the authors should add an arrow in the image and indicate exactly what is going on in each image.

The authors agree with reviewer suggestion and modified the image adding arrows to better indicate what is going on in each stage of image. Moreover, in the revised version of the paper, the author added tags to each image of Figure 1 and modified the figure legend accordingly.

  1. Kindly see the typo in the figure 1 legend. It must be “Histologic Image“ not “Hystologic image”. Figure 2: It schematically represents the classical approach of tissue engineering, but the author indicates this figure in the sentences (line number 763-766) are not suitable. Please check and cite Figure 2 appropriately.

The author agree with reviewer comment and have now cited Figures 1,2 appropriately.

  1. Figures 1, 2, and 3: The present form is not clear enough to publish, better resolution images should be provided.

Images resolution has been improved accordingly to Pharmaceutics guidelines (minimum 1000 pixels width/height, or a resolution of 300 dpi or higher).

  1. Since the major focus of this review is on “New emergent technologies for skin wound care and regeneration”, authors are advised to incorporate a dedicated section to explain the potential role and the status of skin wound care products (which are produced from the discussed emergent technologies in the review) in clinical and pre-clinical studies and rewrite the discussion accordingly.

Skin wound care products and gold-standards procedures were already mentioned and discussed in each sections, a further paragraph will result pleonastic. Moreover, few examples of commercial products are also summarized in the Table 1.

  1. I recommend adding one more figure which schematically explains the types of “New emergent technologies for skin wound care and regeneration”.

As suggested by the Reviewer (Question 2) a systematic summary of new emergent technologies for skin wound care and regeneration has been provided in Table 1. No further figures were included in the manuscript to avoid any overlapping with details collected in Table 1.

  1. Write a separate section on “Future perspectives” must discuss here on the recent development in new emergent technologies for skin wound care and regeneration and outlook of the prospects use in tissue engineering and wound healing applications.

Section Future perspectives was included in the revised version of manuscript.

  1. Please check the Section and sub-section number carefully and correct it, the numbers are repeated in Section 4 and 5. Section 4 previously discusses on “chronic wounds” and the same number given to the manuscript overall “discussion” (see line number 1220), similarly, Section 5 discussed on “skin regeneration process” and the same number given to the manuscript overall “conclusion” (see line number 1264).

Section and sub-section number were carefully revised and corrected.

  1. I would suggest the authors re-organize their manuscript. The information given in Section 5-7 is more appropriate to the title and abstract of the review article. In the present form, the information given in Section 3 & 4 are very elaborate, although I may ask the author to re-write briefly with providing the necessary outcomes from Section 3 and 4.

In revised version, section 3 & 4 were numbered as 2.3. These section were revised and briefly re-writed.

Reviewer 2 Report

In this manuscript "Skin Wound Healing Process and New Emergent Technologies for Skin Wound Care and Regeneration" the authors present a comprehensive overview of cascades of events underpinning wound healing and treatment strategies.

The manuscript presents an interesting analysis of the literature on the topic. The manuscript is well-written, easy to understand, and supported by well-designed themes. However, following major comments need to be addressed;

(i) To make the article less boring and easy for the readers to comprehend, it would be nice to have a schematic representation of various stages involved in wound healing. In fact, I strongly advise that the authors reduce the word counts in order to include nice images. 

(ii) There are numerous short paragraphs that could have flowed better if merged. For example, Line 929-949, 962-963, 964-968, 984-985, etc.

(iii) The authors should seek the help of a native speaker to improve grammar.

Author Response

Pharmaceutics

Manuscript Title: Skin Wound Healing Process and New Emergent Technologies for Skin Wound Care and Regeneration

Authors: Erika Tottoli, Rossella Dorati *, Ida Genta , Enrica Chiesa , Silvia Pisani, Bice Conti

Ms. Ref. No.: pharmaceutics-858731

We are grateful to the Reviewers for their constructive comments and suggestions to improve the manuscript. Our point-by-point responses to the reviewers are presented below:

Reviewer #2

In this manuscript "Skin Wound Healing Process and New Emergent Technologies for Skin Wound Care and Regeneration" the authors present a comprehensive overview of cascades of events underpinning wound healing and treatment strategies.

The manuscript presents an interesting analysis of the literature on the topic. The manuscript is well-written, easy to understand, and supported by well-designed themes. However, following major comments need to be addressed;

(i) To make the article less boring and easy for the readers to comprehend, it would be nice to have a schematic representation of various stages involved in wound healing. In fact, I strongly advise that the authors reduce the word counts in order to include nice images. 

A schematic representation of wound healing stages has been included in the manuscript.

(ii) There are numerous short paragraphs that could have flowed better if merged. For example, Line 929-949, 962-963, 964-968, 984-985, etc.

As suggested by the reviewer short paragraph were revised and in some parts merged.

(iii) The authors should seek the help of a native speaker to improve grammar.

English grammar corrections and text editing were performed.

Reviewer 3 Report

The manuscript entitled "Skin Wound Healing Process and New Emergent Technologies for Skin Wound Care and Regeneration" describes the overall processes of wound healing and, specifically focusing on chronic wounds, introduce the technologies for wound treatment and regeneration. I think the manuscript is written well and the readability is good. Just I feel in most of the parts in the manuscript, the explanations are somewhat too long for a lot of parts. This makes the literature slightly boring to read. Some English corrections or editing may be required. Considering polishing the writing may improve the manuscript, but, overall I think it could be recommended acceptance as the present form. 

Author Response

Pharmaceutics

Manuscript Title: Skin Wound Healing Process and New Emergent Technologies for Skin Wound Care and Regeneration

Authors: Erika Tottoli, Rossella Dorati *, Ida Genta , Enrica Chiesa , Silvia Pisani, Bice Conti

Ms. Ref. No.: pharmaceutics-858731

We are grateful to the Reviewers for their constructive comments and suggestions to improve the manuscript. Our point-by-point responses to the reviewers are presented below:

Reviewer #3

The manuscript entitled "Skin Wound Healing Process and New Emergent Technologies for Skin Wound Care and Regeneration" describes the overall processes of wound healing and, specifically focusing on chronic wounds, introduce the technologies for wound treatment and regeneration. I think the manuscript is written well and the readability is good. Just I feel in most of the parts in the manuscript, the explanations are somewhat too long for a lot of parts. This makes the literature slightly boring to read. Some English corrections or editing may be required. Considering polishing the writing may improve the manuscript, but, overall I think it could be recommended acceptance as the present form.

As suggested by the Reviewer the explanations were revised and made more fluent, english grammar corrections and text editing were performed.

Round 2

Reviewer 1 Report

  1. There are some typos in the article, I request the author to check the manuscript thoroughly for the typos.

Eg.

  • Line number 281, section 2.5, it must be “Scarring” not “Sacrring”,
  • line number 190, “Physiological” not “Phisiological”
  • line number 376, “accepted” not “acceptaed”
  • line number 487, “protecting” not “ptotecting”
  • line number 897, “poly lactic-…..” not “poly lactice-…..”